# Reaching a cell monolayer at the end of hiPSC differentiation enhances neural crest lineage commitment

**Filipa M. Duarte**[1,2]*, **Jeroen van de Peppel**[2], **Irene M.J. Mathijssen**[1], **Johan W. van Neck**[1]

**1** Department of Plastic and Reconstructive Surgery and Hand Surgery, Erasmus University Medical Center, Rotterdam, The Netherlands, **2** Department of Internal Medicine, Erasmus University Medical Center, Rotterdam, The Netherlands

* ana_filipa.duarte@hotmail.com

## Abstract

Neural crest stem cells (NCSCs) compose a highly migratory, multipotent, stem cell population arising from the neural plate border of the embryonic ectoderm. Investigating the development of NCSCs is critical in understanding both embryonic development and abnormal events that underlie neurocristopathies. Suggested seeding densities in *in vitro* human induced pluripotent stem cells (hiPSCs) differentiation protocols, varying between 10,000 cells/cm$^2$ and 200,000 cells/cm$^2$, demonstrate a lack of consensus on the optimal conditions to obtain NCSCs. Aiming to maximize the differentiation efficiency of hiPSCs towards the NCSCs lineage, we investigated the effect of the initial seeding density on NCSCs lineage commitment, both in fibroblast- and human peripheral blood mononuclear cell (PBMC)-derived hiPSCs. Cultures were characterized with gene and protein expression analysis assessing stemness (*OCT3/4* and *NANOG*), neural crest identity (*SNAI2* and *SOX10*) and neuroectoderm identity (*PAX6* and *SOX1*). We demonstrate that reaching a confluent monolayer of cells by the end of the differentiating protocol is crucial to obtaining NCSCs from hiPSCs. To achieve this, our results indicated 17,000 cells/cm$^2$ is the optimal initial seeding density. Under this protocol, a confluent monolayer was reached after 8 days of differentiation and an average of 89% SOX10 positive cells were obtained. The fold change of *SNAI2* and *SOX10* expression was 11-fold and 17-fold higher, respectively, in cultures seeded with 17,000 cells/cm$^2$, compared to the highest tested density of 200,000 cells/cm$^2$. In contrast, seeding 200,000 cells/cm$^2$ induced neuroectoderm-like cells, confirmed by an average of 45% of cells marking positive for PAX6. With this work, we demonstrate the importance of achieving cellular confluency during NCSCs differentiation.

**Data availability statement:** 1) Relevant data are within the manuscript and its Supporting information files 2) Additionally, all gene expression data files are available from the OSF database (accession link: https://osf.io/9fmb4/?view_only=c0cdb683481a476bb-f7975ab0bdd9d7c).

**Funding:** European Union's Horizon 2020 research and innovation program under Marie Skłodowska-Curie (grant agreement No 860635).

**Competing interests:** The authors declare no competing interests.

## Introduction

Human induced pluripotent stem cells (hiPSCs) have the ability to differentiate into the specialized cell types that constitute the three main germ layers: ectoderm, mesoderm and, endoderm [1]. The timeline and phenotypic outcome of cellular differentiation depend on a variety of molecular signals that originate both from within the cell and from the extracellular environment [2,3].

Maintenance of pluripotency and self-renewal is controlled by different transcription factors such as *OCT4, SOX2, KLF4, c-MYC, NANOG, LIN28, TCF3 and KLF4* [4]. Additionally, mechanical signals, produced as a consequence of intracellularly generated and externally applied forces, can also influence differentiation and cell fate commitment [5].

Neural crest stem cells (NCSCs) are a highly migratory cell population that is unique to vertebrates [6]. NCSCs arise from the embryonic ectoderm, delaminate from the neural plate border, and migrate across the embryo to give rise to a heterogeneity of cell types such as craniofacial bone, cartilage cells, smooth muscle cells, melanocytes, peripheral and enteric neurons, and glia [7].

During NCSCs development, a strict cascade of events regulates their maintenance, epithelial-mesenchymal transition, and migration. Initially, *FOXD3, SNAI1/2, SOX10, TFAP2A, TCOF, ALX1, and NLZ1/2,* promote NCSCs identity and maintain pluripotency. Then, *TCOF, NLZ1/2*, and *SNA1/2*, along with *ETS1*, *MAF*, and *ZEB2* promote epithelial-mesenchymal transition, which is followed by delamination and migration of NCSCs triggered by the CHARGE-associated genes, *CHD7, KMD6A, BRG1*, and *KMT2D* [8].

Improper neural crest (NC) development results in a class of pathologies designated as neurocristopathies [9]. Treacher Collins syndrome, Pierre Robin sequence, Waardenburg–Shah syndrome, CHARGE syndrome and DiGeorge syndrome as well as neuroblastoma are examples of neurocristopathies that affect thousands of patients every year [10]. Investigating the development of NCSCs is crucial for understanding the etiology of neurocristopathies and for developing novel therapeutic strategies.

Deriving NCSCs from hiPSCs and embryonic stem cells (ESCs) *in vitro* has become a powerful strategy for dissecting the stages of neural crest development [11]. A number of *in vitro* protocols to obtain NCSCs from hiPSCs have been published [12–19]. In these protocols, initial seeding densities range from 10,000–200,000 cells/cm$^2$ which demonstrates a lack of consensus on the most appropriate culture conditions to maximize neural crest differentiation efficiency.

In this report, we investigate the optimal hiPSC seeding density that maximizes the efficiency of obtaining NCSCs from hiPSCs in an 8-day NC differentiation protocol, testing a range of densities from 8,500 cells/cm$^2$ to 200,000 cells/cm$^2$. We demonstrate that achieving a confluent monolayer of cells by the end of the hiPSCs to NC differentiation protocol is critical for inducing the NC cell profile. In contrast, when a multi-layered culture is present at the end of the differentiation protocol, neuroectoderm PAX6 positive populations emerge.

## Materials and methods

### hiPSCs lines

All lines were derived from healthy individuals and reprogrammed into hiPSCs with the CytoTune-iPS 2.0 Reprogramming kit (ThermoFisher Cat#A16517) and reported in the hPSCreg database: EMCi94-A (fibroblast-derived, donor 1), EMCi230-A (PBMC-derived, donor 2), EMCi238-A (PBMC-derived, donor 3) [20,21].

### hiPSCs maintenance

hiPSCs were cultured on 10 cm culture plates (Greiner Cat#664160) coated with a Matrigel substrate (Corning Cat#356278) in mTeSR plus medium (STEMCELL Technologies Cat#100–0276). All hiPSCs lines were passaged at 70–80% confluency using ReLeSR (STEMCELL Technologies Cat#100–0484) to detach the cells from the plate surface: after washing the hiPSCs cultures with PBS, ReLeSR was applied and immediately removed, followed by an incubation of 5 minutes at 37°C. Subsequently, cells were collected by resuspending in mTeSR plus and plated at a 1:10 dilution in aggregates of 2–4 cells for further expansion.

### Neural crest induction

StemDiff™ Neural crest differentiation kit (STEMCELL Technologies Cat#08610) was used according to the manufacturer's guidelines, while varying cell seeding densities. Cells were counted using a Countess 3 automated cell counter (Thermo Fisher Scientific Inc.) and seeded at the varying seeding densities in Matrigel (Corning Cat#356278) coated 6 and 12-well plates (Greiner Cat#665180). Medium changes happened daily during the NC differentiation timeline.

### Bright field and fluorescent imaging

Pictures of the colonies were collected using a ZEISS Axio Observer Z1 PALM fluorescent microscope.

### Gene expression analysis

To determine the phenotypic identity of the cells before and after undergoing NC differentiation, Real-Time quantitative PCR (RT-qPCR) analysis was conducted. RNA was isolated with TRIzol Reagent (Thermo Fisher Scientific Inc. Cat#15596−026) following the manufacturer's guidelines. Then, cDNA was synthesized using a cDNA synthesis kit (Thermo Fisher Scientific Inc. Cat#K1622). To conduct gene expression analysis, 3 biological replicates for each condition were assessed on a QuantStudio 12K Flex cycler (Thermo Fisher Scientific Inc.) using GoTaq qPCR Master Mix (Promega Cat#A6002). CT-values were normalized to GAPDH, and gene expression values were compared using the ΔΔCT-method [22]. The primers used for the gene expression measurements can be found in the S1 Table.

### Immunocytochemistry

hiPSCs were seeded in µ-Slide 4 Wells (Ibidi Cat#80426) that were coated with Matrigel. After 8 days of NC differentiation (or in mTeSR plus media in case of control hiPSCs conditions) the cultures were fixed with 4% PFA for 5 minutes at room temperature followed by three washes with PBS. Next, the slides were incubated in blocking solution (5% BSA, 0.1% Triton X-100, 0.02% Tween20 in PBS) for 30 minutes followed by overnight incubation with primary antibodies directed against human PAX6 (R&D Systems Cat#AF8150, 1:500), SOX10 (R&D Systems Cat#MAB2864, 1:500) or OCT3/4 (Abcam Cat#ab19857, 1:250). The following day, the wells were washed three times with PBS and incubated for 30 minutes with the respective secondary antibodies: Alexa Fluor®594 donkey anti-sheep (ThermoFisher Cat#A-11016, 1:500),

 

Alexa Fluor®488 donkey ant-rabbit (ThermoFisher Cat#R-37118, 1:500) and Alexa Fluor®488 donkey anti-mouse (ThermoFisher Cat#A-21202, 1:500). After washing with PBS, the wells were covered with a mounting gel containing DAPI (Vectashield Cat#H-1200–10) and fluorescent images were captured at 10x magnification.

## Flow cytometry analysis

Wells containing hiPSCs or NCSCs were washed once with PBS and were detached by adding TrypleE (Thermo Fisher Scientific Inc. Cat#25200056) for 10 minutes at 37°C. Once detached, cells were mechanically dissociated into a single cell suspension by shear force through a 1 mL pipette and subsequently centrifuged at 300g for 5 minutes. To fix the cells, the pellets were resuspended in fixation buffer (BDbiosciences Cat#554655) and incubated on ice for 15 minutes at 4°C. Subsequently, three washes with Perm/Wash buffer (BDbiosciences Cat#554723) were performed followed by an incubation step of 15 minutes in Perm/Wash buffer at 4°C. The buffer was then removed by centrifuging the cells at 300g for 5 minutes. Subsequently, antibodies directed against human PAX6 PerCP-Cy™5.5 (BDbiosciences Cat#562388, 1:50), OCT3/4 Alexa Fluor®488 (BDbiosciences Cat#560253, 1:50) or SOX10 (R&D Systems Cat#MAB2864, 1:50) were added for 30 minutes at room temperature. All antibodies were diluted in Staining buffer (BDbiosciences Cat#554656). Cells were washed three times with Perm/Wash buffer and resuspended in a Staining buffer. To target SOX10, secondary staining was carried out with Alexa Fluor®488 donkey anti-mouse (ThermoFisher Cat#R-37118, 1:50) for a period of 30 minutes at room temperature. Finally, each cell's phenotypic identity was determined with a BD Accurri C6 flow cytometer.

## Statistical analysis

A Mann–Whitney U test was used to assess the effects of cell seeding density and differentiation day on marker gene expression. Statistical significance was set at $p < 0.05$ (*), $p < 0.01$ (**), and $p < 0.001$ (***). Analyses were performed using Python.

## Results

### Fibroblast-derived hiPSCs seeded at 17,000 cells/cm$^2$ reach confluency in 8 days, maximizing neural crest gene expression

NCSCs differentiation protocols from hiPSCs show differences in multiple, partly intertwined, factors that exert an influence on achieving NC identity such as seeding densities, duration of differentiation, coating matrix, and growth factors (Table 1). Among these, the most significant difference we observed was the considerable variation in the initial cell seeding densities. To study the effects of the initial seeding density on NCSC differentiation and maximize hiPSC to NCSC differentiation efficiency, we seeded fibroblast-derived hiPSCs from donor 1, at 8,500, 17,000, 34,000, 63,000, 90,000,

Table 1. Summary of published Neural crest differentiation protocols. Variability in methodologies to achieve neural crest stem cell identity from hiPSCs across published differentiation protocols.

| Initial seeding density (cells/cm$^2$) | Length of Differentiation | NCSCs inductive compounds | Coating | Reference |
|---|---|---|---|---|
| 10,000 | 7 days | N2 supplement, CHIR99021, SB431542, BMP4, DMH-1 | Vitronectin | Mehler, Burns et al. 2020 [12] |
| 20,000 | 5 days | CHIR99021 | Matrigel | Leung et al. 2016 [13] |
| 30,000 | 21 days | Normocin, bFGF, SB431542, CHIR99021 | Matrigel | Kobayashi, Musso et al. 2020 [14] |
| 100,000 | Not described | Activin A inhibitor, GSK3 inhibitor, SB 431542 | Not described | Menéndez et al. 2011 [16] |
| 200,000 | 6 days | STEMdiff™ Neural Crest Differentiation Kit | Matrigel | Elliott et al.2023 [17] - https://www.stemcell.com/products/stemdiff-neural-crest-differentiation-kit.html |

and 200,000 cells/cm$^2$. These were differentiated for 8 days using the STEMdiff™ Neural Crest Differentiation Kit. Subsequently, we investigated the effect on gene expression of stem cell-, NCSC-, and neuroectoderm lineage markers.

In contrast to hiPSCs cultures in maintenance media, NC differentiated hiPSCs did not display contact inhibition when cultured in NCSC differentiation medium. When the initial seeding density surpassed 17,000 cells/cm$^2$, a multi-layered cell culture was observed on day 8 (Fig 1A). We determined the gene expression profiles of the stem cell markers *OCT3/4* and *NANOG* every other day for each cell seeding density and observed that all densities abandoned their pluripotency state after 2 days of NC differentiation (Fig 1B).

Since the NC arises from the early ectoderm plate border, the most frequently observed non-NCSCs cell types during NC differentiation are expected to be of ectodermal origin [15]. Therefore, the expression of the NC markers *SNAI2* and *SOX10* (Fig 1C), as well as the expression of the ectoderm markers *PAX6* and *SOX1* (Fig 1D), were determined during differentiation.

The NC markers *SNAI2* and *SOX10* were predominantly expressed at the lowest cell seeding densities and peaked at 17,000 cells/cm$^2$. *SNAI2* and *SOX10* expression were, respectively, 11 and 17 times higher after 8 days of NC differentiation in cultures seeded at 17,000 cells/cm$^2$ when compared to cultures seeded with 200,000 cells/cm$^2$ (Fig 1C).

The ectoderm markers *PAX6* and *SOX1* followed the opposite trend: low expression at low seeding densities and the highest expression at the highest cell seeding density. *PAX6* and *SOX1* expression were, respectively, 191 and 15 times lower in cultures seeded at 17,000 cells/cm$^2$ when compared to cultures seeded with 200,000 cells/cm$^2$ (Fig 1D).

At day 8 of NCSCs differentiation, the expression of *SNAI2* doubled from day 6, in cultures seeded with 17,000 cells/cm$^2$ (Fig 1C), whereas the gene expression of the neuroectoderm markers *PAX6* and *SOX1* only slightly differed on day 6, compared to day 8 of differentiation (Fig 1D).

In summary, our data demonstrated that reduced seeding densities promote NC identity within an 8-day induction protocol.

## PBMC-derived hiPSCs optimally differentiate into the NC lineage after 8 days when seeded at 17,000 cells/cm$^2$

To verify that reduced cell seeding densities enhance the quantity of NC differentiated cells, as noted in our experiments with hiPSCs from donor 1, we additionally validated our NCSCs differentiation protocol utilizing two distinct hiPSCs lines derived from peripheral blood mononuclear cells (PBMCs) (donor 2 and donor 3). To ensure consistency across experiments and validate the reproducibility of our observations, we designed the experimental setup to assess identical conditions, thereby minimizing any potential variability and confirming the reliability of the results observed in donor 1. Both lines were immuno-stained for the stemness marker OCT3/4, which showed widespread expression throughout the cell cultures, as demonstrated by the overlap of DAPI and OCT3/4 (S1 Fig). After 8 days of NC differentiation induction, both lines revealed a significant decrease in *OCT3/4* and *NANOG* gene expression, confirming their ability to exit the pluripotent state. *NANOG* expression decreased 2–16 fold when cells were seeded at densities of 17,000 cells/cm$^2$ and 200,000 cells/cm$^2$. Similarly, *OCT3/4* expression was significantly reduced by 17–111 fold (S1 Fig). Together, these results confirm the initial hiPSCs identity and demonstrate that both lines effectively abandon the pluripotent state under both initial seeding conditions.

NC induction of hiPSCs seeded at 200,000 cells/cm$^2$ reached confluency as early as Day 1 of the differentiation period (Figs 2A and 2B). Consistent with our observations in fibroblast-derived hiPSCs from donor 1, both PBMC derived hiPSCs lines did not display contact inhibition upon NC differentiation, which resulted in super confluency and the formation of multi-layered cell cultures over subsequent days. In contrast, cultures seeded at 17,000 cells/cm$^2$ reached a confluent monolayer at the end of the NC differentiation period (Day 8) (Figs 2A and 2B).

In donor 2, the expression of *SNAI2* and *SOX10* were 8 and 7 times higher, respectively, in cultures seeded at 17,000 cells/cm$^2$ compared to 200,000 cells/cm$^2$. In contrast, *PAX6* and *SOX1* expression were 89 and 17 times lower in cultures seeded at 17,000 cells/cm$^2$ compared to 200,000 cells/cm$^2$ (Fig 2C). Cells originating from donor 3 exhibited a similar

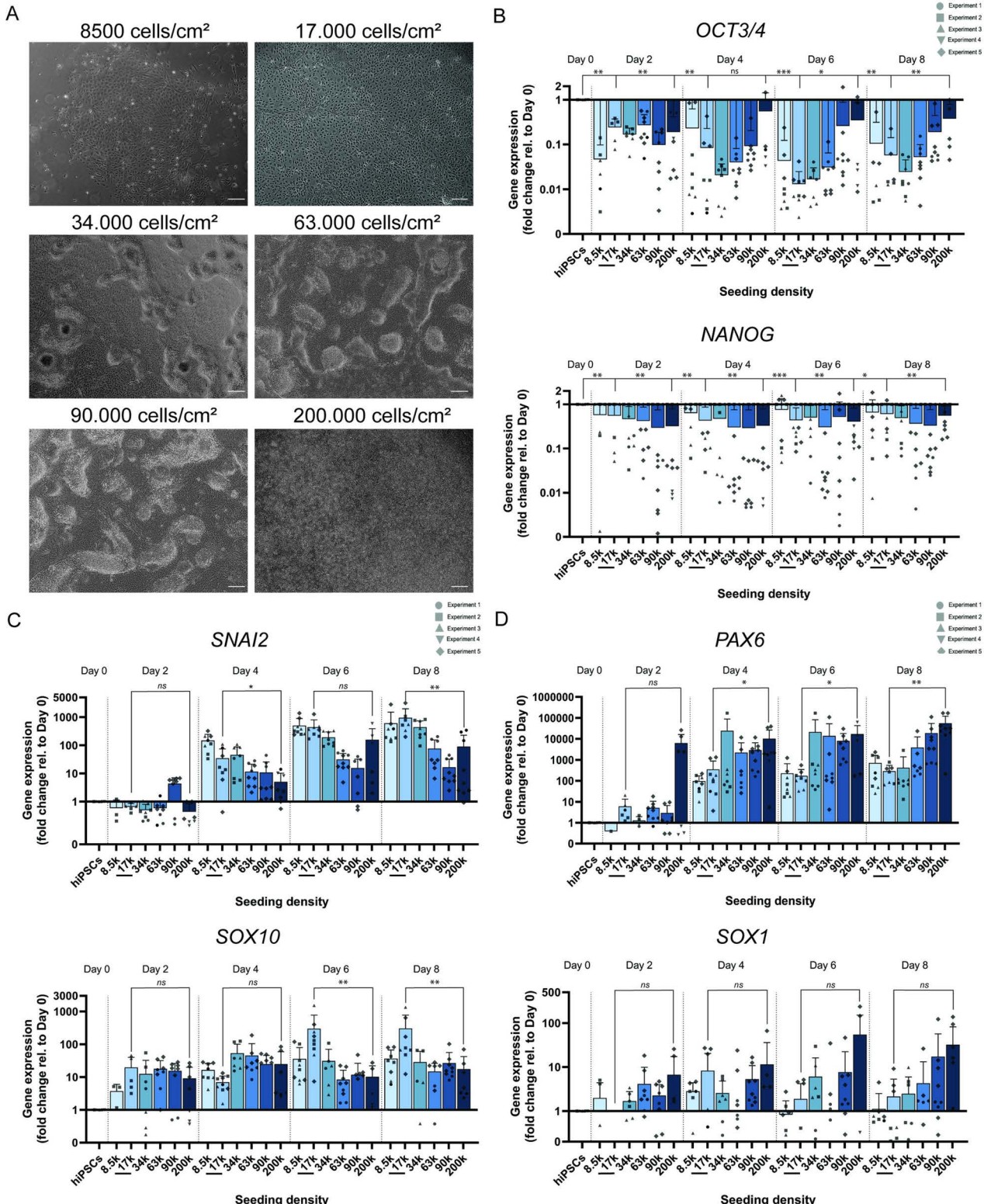

**Fig 1. Cell seeding density determines cell fate during neural crest induction of fibroblast-derived hiPSCs. (A)** Bright-field images of hiPSCs cultures from donor 1 at different seeding densities (8,500, 17,000, 34,000, 63,000, 90,000, and 200,000 cell/cm²), after 8 days of neural crest differentiation. Scale bars = 100 µm. Gene expression of stemness markers *OCT3/4* and *NANOG* indicate loss of pluripotency after NC induction **(B)**. Neural crest

identity markers *SNAI2* and *SOX10* (C) are highly expressed at lower seeding densities and neuroectoderm markers *PAX6* and *SOX1* are expressed at higher seeding densities **(D)**. Gene expression was assessed before (in hiPSCs) and after 2,4,6, or 8 days of neural crest differentiation at the indicated initial cell seeding densities. Fold changes were calculated relative to expression in hiPSCs (D0). Error bars represent the standard deviation. *: p-value < 0.05, **: p-value < 0.01, ***: p-value < 0.001, ns: not significant.

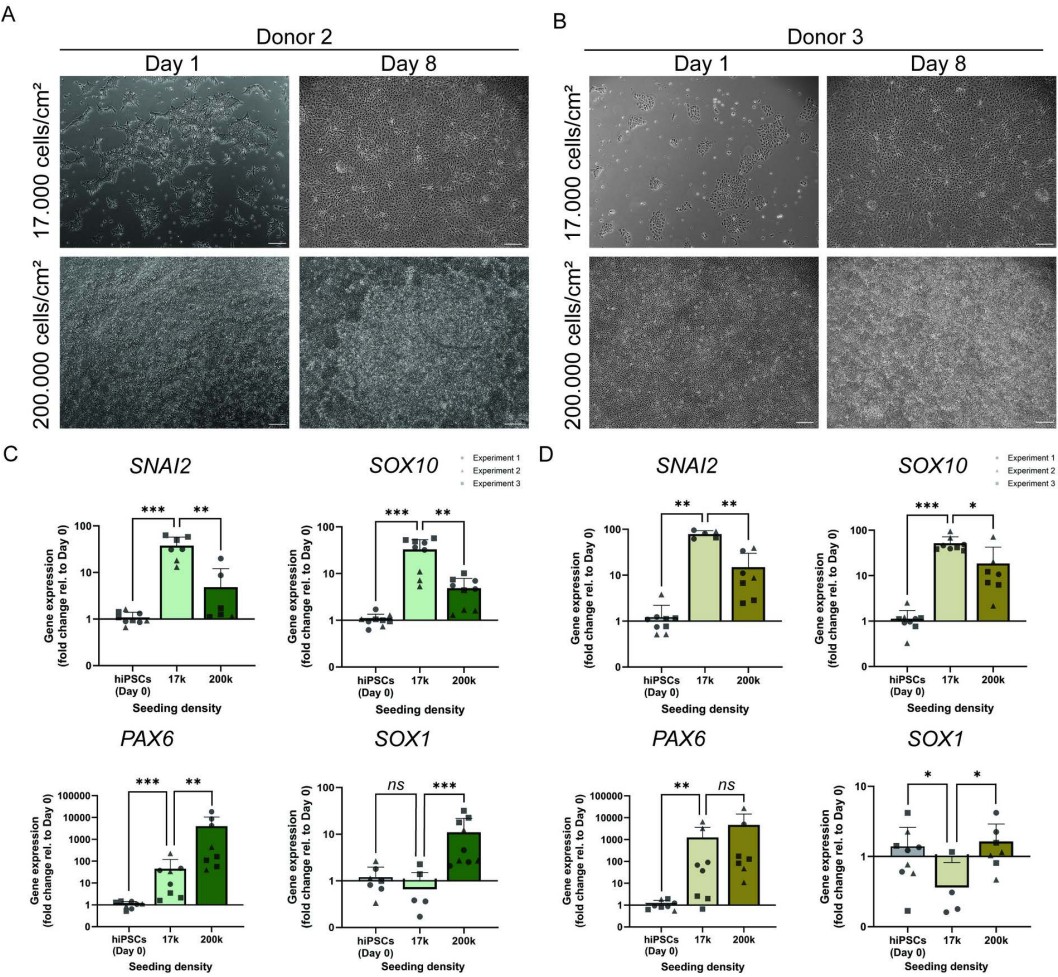

**Fig 2. Increased differentiation efficiency of PBMC-derived hiPSCs into neural crest stem cells at lower cell seeding densities.** Bright field images of donor 2 (A) and donor 3 **(B)** PBMC-derived hiPSCs at Day 1, and the last day of the neural crest differentiation period (Day 8), seeded at either 17,000 cells/cm$^2$ or 200,000 cells/cm$^2$. Cultures seeded at 17,000 cells/cm$^2$ formed a confluent monolayer by Day 8, while cultures seeded at 200,000 cells/cm$^2$ formed multi-layered cell cultures. Scale bars = 100 μm. Gene expression analysis of neural crest genetic markers *SNAI2* and *SOX10* indicates higher expression at lower seeding densities, whereas neuroectoderm genetic markers *PAX6* and *SOX1* show increased expression at higher densities. Cultures derived from donor 2 are marked in green (C) and those of donor 3 in brown **(D)**. Fold changes are calculated relative to the expression in hiPSCs (at Day 0). Error bars represent the standard deviation (n = 3). *: p-value < 0.05, **: p-value < 0.01, ***: p-value < 0.001, ns: not significant.

trend: *SNAI2* and *SOX10* expression were 5 and 3 times higher, respectively, in cultures seeded with 17,000 cells/cm$^2$ than with 200,000 cells/cm$^2$. Conversely, the expression of *PAX6* and *SOX1* were 4 and 5 times lower, respectively, in cultures seeded with 17,000 cells/cm$^2$ compared to 200,000 cells/cm$^2$ (Fig 2D). This data corroborates our previous observations and demonstrates that two unrelated PBMC-derived hiPSCs lines also differentiate more efficiently to NC cells at lower initial cell seeding densities.

## NCSCs thrive in confluent monolayer cultures while neuroectoderm-like cells emerge in regions of superconfluency

To assess the protein expression profile of the hiPSCs after 8 days of NCSCs induction, we immuno-stained cultures seeded at 17,000 cells/cm² and 200,000 cells/cm² for PAX6 and SOX10. SOX10 expression was scarce in cultures seeded at 200,000 cells/cm². In contrast, cultures seeded at 17,000 cells/cm² exhibited uniform SOX10 expression across the confluent monolayer of cells (Fig 3A).

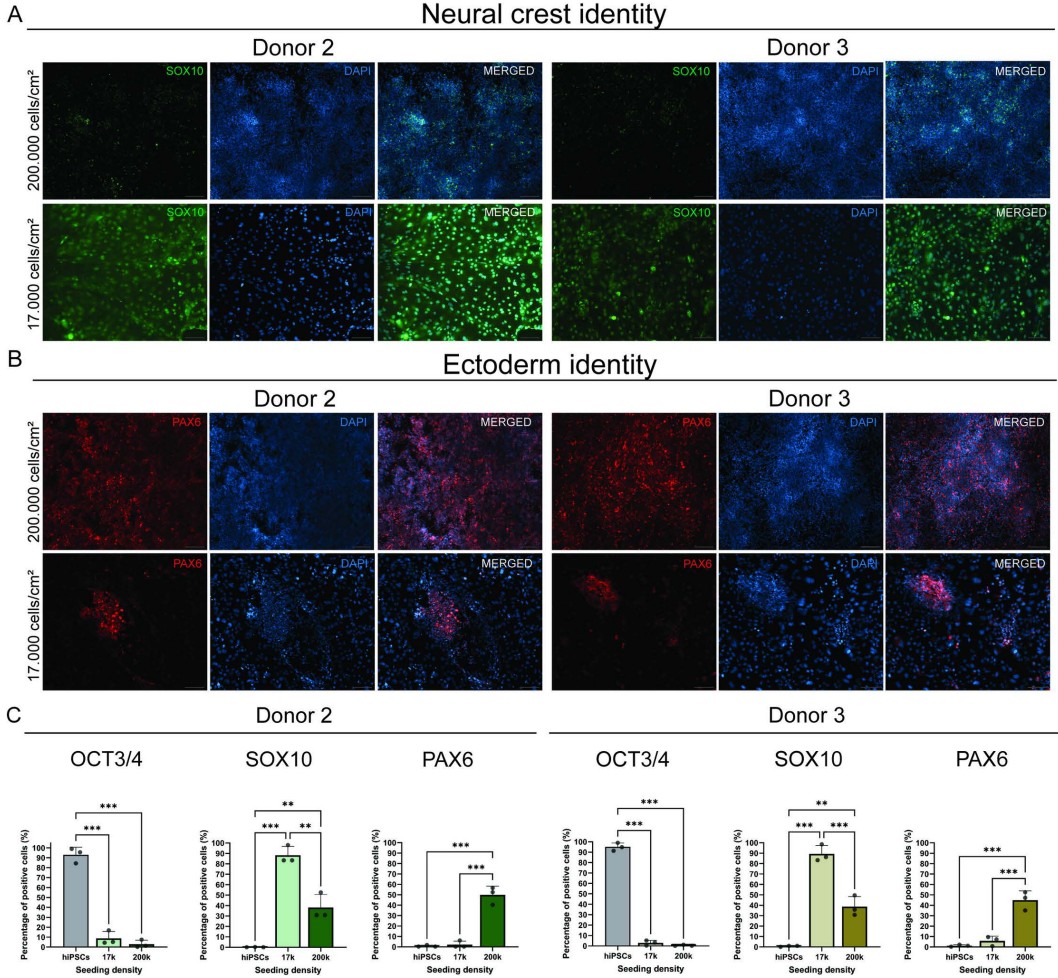

**Fig 3. Neural crest differentiated PBMC-derived hiPSCs express neural crest markers in areas with a monolayer of cells and neuroectoderm markers in areas of multi-cell layers. (A)** After an 8 days neural crest differentiation protocol, in donors 2 and 3, SOX10 protein expression was detected in monolayer regions of cultures seeded at 17,000 cells/cm², while it was absent in cultures seeded at 200,000 cells/cm². **(B)** Following neural crest differentiation, PAX6 protein expression was detected in areas with multiple layers of cells in cultures seeded at 17,000 cells/cm² and was more abundant in cultures seeded at 200,000 cells/cm². **(C)** Flow cytometry analysis of hiPSCs and neural crest differentiated cells seeded at 17,000 cells/cm² and 200,000 cells/cm². Samples were immuno-stained for the stemness protein marker OCT3/4, the neural crest marker SOX10, and the neuroectoderm protein marker PAX6. OCT3/4 is downregulated by the end of the differentiation period at both 17,000 cells/cm² and 200,000 cells/cm² initial cell seeding densities. SOX10 shows high expression under the 17,000 cells/cm² condition, in both donors, while PAX6 is strikingly expressed only under the 200,000 cells/cm² condition. Differentiated NCSCs from donor 2 and donor 3 are represented in green and brown, respectively and hiPSCs are represented in grey. Error bars represent standard deviation (n=3). *: p-value < 0.05, **: p-value < 0.01, ***: p-value < 0.001.

Oppositely, PAX6 was more abundantly expressed in cultures plated at 200,000 cells/cm$^2$ and was detected only in highly dense cell clusters or multilayered areas of cell cultures plated at 17,000 cells/cm$^2$ (Fig 3B).

To evaluate the quantitative efficiency of the NC differentiation protocol, we performed flow cytometry analysis on both PBMC-derived hiPSCs donor lines. Consistent with our gene expression and immunofluorescent results, all cells lost pluripotency during NC differentiation. Cultures initially seeded with 17,000 cells/cm$^2$ showed an average of 88.2% and 89.4% of cells positive for SOX10 in donor 2 and donor 3, respectively. In contrast, cultures seeded at 200,000 cells/cm$^2$ showed lower percentages of SOX10 positive cells, with 38% and 38.7% marking positive in donor 2 and donor 3, respectively. The percentage of PAX6-positive cells was higher under the 200,000 cells/cm$^2$ seeding condition, measuring 49.8% and 44.9% positive cells in donor 2 and donor 3, respectively, compared to the low-density cultures seeded at 17,000 cells/cm$^2$, where only 2.2% and 5.9% PAX6-positive cells were found in donor 2 and donor 3, respectively. (Fig 3C).

## Discussion

Published protocols for differentiating neural crest (NC) cells from human induced pluripotent stem cells (hiPSCs) *in vitro* report initial seeding densities that range from 10,000–200,000 cells/cm$^2$, which highlights a lack of consensus on the optimal strategy for generating neural crest stem cells (NCSCs) *in vitro* [12–19].

This report investigates how the initial cell seeding density influences the efficiency of hiPSC to NC differentiation. We demonstrate that, by the end of the differentiation period, SOX10-positive NC cells are predominantly produced in the monolayer regions of the cell cultures, whereas the majority of PAX6-positive neuroectodermal cells predominantly are found in superconfluent, multi-layered areas. Such observations were replicated across all lines that were tested (one fibroblast-derived hiPSCs line and two PBMC-derived hiPSCs lines). Optimal SOX10 expression is achieved by seeding 17,000 hiPSCs/cm$^2$ and a differentiation period of 8 days in NC inductive media. However, due to cell line variability, it is essential to recognize the need to evaluate the optimal culture and seeding conditions, that will give a confluent monolayer of cells at the end of the differentiation protocol, for each line.

Etoc et al., 2016, utilizing pluripotent embryonic stem cells (ESCs), described the mechanism by which ESCs regulate identity fate. Cells at the edge of the colony express apical TGF-β receptors and remain responsive to TGF-β Ligands. In contrast, cells situated further from the edge relocate their TGF-β receptors to the lateral side, which results in an insensitivity to TGF-β signalling [23]. We suggest that a similar mechanism might contribute to the low efficiency of NC differentiation that we observed in multi-layered cultures: restricted access to factors or signaling events in superconfluent cell areas may impede effective NC differentiation.

Upon evaluating eight non-commercial NC differentiation protocols, we observed that all included CHIR99021, a GSK3 inhibitor (GSK3 phosphorylates β-catenin, leading to its degradation and consequently inhibiting the Wnt/β-catenin pathway that is essential for NCC development); SB431542, which is a TGF-β/Activin/Nodal signalling pathways inhibitor, involved in mesoderm and endoderm formation. Five out of the eight protocols used both factors synergistically [13,15,19,24–28].

The finding that our cultures, when seeded with 34,000 cells/cm$^2$ or higher densities, show areas of superconfluency that are positive for PAX6, seems to reinforce the notion that the absence of GSK3 inhibition factors or TGF-β/Activin/Nodal inhibition factors impedes the acquisition of NC fate, resulting in a predominance of neuroectoderm differentiation.

In summary, as tested in fibroblast and PBMC-derived hiPSCs lines, to enhance the yield of NC following hiPSC to NC differentiation it is essential to obtain a confluent monolayer of cells by the conclusion of the differentiation period. Our protocol delivers an efficiency of approximately 90% in producing SOX10 positive NC cells, which significantly reduces the need for FACS-assisted purification. High NC yields will facilitate exploring the fundamental biology of NC cells, disease modeling of neurocristopathies, and strategies in regenerative medicine.

## Supporting information

**S1 Table: Sequence of the primers used in gene expression analysis.**
(DOCX)

**S1 Fig. PBMC-derived hiPSCs undergoing neural crest differentiation successfully abandon the pluripotent state.**
Immunocytochemistry analysis of OCT3/4 expression in PBMC-derived hiPSCs from donor 2 (A) and donor 3 (B) confirms extensive stemness marker expression. Scale bars = 100 μm. Gene expression analysis of *OCT3/4* and *NANOG* in donor 2 (C) and donor 3 (D), show loss of pluripotency after NC induction and was assessed on undifferentiated cells (Day 0, grey bars) and after 8 days of neural crest differentiation, with a 17,000 cells/cm$^2$ and 200,000 cells/cm$^2$ initial seeding density. Differentiated NCSCs from donor 2 and donor 3 are represented in green and brown, respectively. Fold changes were calculated relative to the expression levels in hiPSCs (at Day 0). Error bars represent standard deviation (n = 3). *: p-value < 0.05, **: p-value < 0.01, ***: p-value < 0.001, ns: not significant.
(TIF)

## Acknowledgments

We thank the Erasmus MC iPS core facility for their contributions to this research. Specifically, we acknowledge Mehrnaz Ghazvini, PhD; Lieke Dons, and Tracy Li for assisting with the cell culture experimental setup, and for their invaluable expertise.

## Author contributions

**Conceptualization:** Filipa M. Duarte, Jeroen van de Peppel, Johan W. van Neck.

**Data curation:** Filipa M. Duarte, Johan W. van Neck.

**Formal analysis:** Filipa M. Duarte.

**Funding acquisition:** Jeroen van de Peppel, Irene M.J. Mathijssen, Johan W. van Neck.

**Investigation:** Filipa M. Duarte, Jeroen van de Peppel.

**Methodology:** Filipa M. Duarte, Jeroen van de Peppel, Johan W. van Neck.

**Project administration:** Filipa M. Duarte, Irene M.J. Mathijssen, Johan W. van Neck.

**Resources:** Irene M.J. Mathijssen, Johan W. van Neck.

**Software:** Jeroen van de Peppel.

**Supervision:** Jeroen van de Peppel, Irene M.J. Mathijssen, Johan W. van Neck.

**Validation:** Filipa M. Duarte, Jeroen van de Peppel.

**Writing – original draft:** Filipa M. Duarte.

**Writing – review & editing:** Filipa M. Duarte, Johan W. van Neck.

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
