## [Decision Letter · Decision Letter 0]

5 May 2025

PONE-D-25-17819Reaching a cell monolayer at the end of hiPSC differentiation enhances neural crest lineage commitment.PLOS ONE

Dear Dr. Duarte,

Thank you for submitting your manuscript to PLOS ONE. After careful consideration, we feel that it has merit but does not fully meet PLOS ONE’s publication criteria as it currently stands. Therefore, we invite you to submit a revised version of the manuscript that addresses the points raised during the review process.

On reading the manuscript, I agree with the comments made by reviewer 1, which you should address explicitly.  There are a number of awkward phrasings throughout the manuscript, together with the repeat of a phrase in

Line 65, that you should correct. 

We look forward to receiving your revised manuscript.

Kind regards,

Michael Klymkowsky, Ph.D.

Academic Editor

PLOS ONE

“European Union’s Horizon 2020 research and innovation program under Marie Skłodowska-Curie (grant agreement No 860635)”

“We thank the Erasmus iPS core facility for their contributions to this research. Specifically, we acknowledge Mehrnaz Ghazvini, PhD; Lieke Dons and Tracy Li for assisting with the cell culture experimental setup, and for their invaluable expertise.  We also wish to thank the European Union’s Horizon 2020 research and innovation program under Marie Skłodowska-Curie (grant agreement No 860635) for their financial support. The results above reflect only the authors’ view, and the Research European Agency is not responsible for any use that may be made of the information it contains.”

“European Union’s Horizon 2020 research and innovation program under Marie Skłodowska-Curie (grant agreement No 860635)”

5. Please amend the manuscript submission data (via Edit Submission) to include author Filipa M. Duarte.

6. Please amend your authorship list in your manuscript file to include author Ana Duarte.

Reviewers' comments:

Reviewer's Responses to Questions

**Comments to the Author**

1. Is the manuscript technically sound, and do the data support the conclusions?

Reviewer #1: Yes

2. Has the statistical analysis been performed appropriately and rigorously? 

Reviewer #1: No

3. Have the authors made all data underlying the findings in their manuscript fully available?

Reviewer #1: Yes

4. Is the manuscript presented in an intelligible fashion and written in standard English?

Reviewer #1: No

5. Review Comments to the Author

Reviewer #1: Duarte et al studied the effect of initial seeding density on the differentiation of human induced pluripotent stem cells (hiPSCs) into neural crest stem cells (NCSCs). They found that achieving cellular confluency by the end of the 8-day differentiation protocol was essential for efficient NCSC generation.

- In Table 1 of the paper, numerous variables are described, including days of differentiation, cell number, coating matrix, and differentiation cocktails. Do the authors believe that cell number alone is the key factor influencing NCSC differentiation, or have the potential contributions of the other variables been considered?

- Please include statistical analysis for the qPCR data presented in Figure 1 and Figure 3 to support the reported differences in gene expression.

- For Figures 1C and 1D, the 17K condition could be highlighted more clearly to aid reader interpretation—perhaps by using connecting lines or another visual cue. Just a suggestion

- Figure 1 was performed using cells from a single donor, whereas Figure 2 and on includes PBMC-derived hiPSCs from two donors. Could the authors clarify why a second donor was not included in Figure 1 to strengthen the robustness and reproducibility of the findings?

- It is unclear what additional insight Figure 2 provides, as the downregulation of OCT4 and NANOG during differentiation is expected regardless of initial cell confluency. This trend was already demonstrated in Figure 1 using fibroblast-derived hiPSCs. Could the authors clarify the specific purpose of Figure 2?

- Line 237 – there is a repetition on ‘confluency’

Line 241 – ‘ain’

- In the legend for Figure 3, consider revising the title to: "Increased differentiation efficiency of PBMC-derived hiPSCs into neural crest stem cells at lower cell seeding densities." The same suggestion applies to the legend for Figure 4 to ensure clarity and consistency regarding the cell source.

6. PLOS authors have the option to publish the peer review history of their article (what does this mean? ). If published, this will include your full peer review and any attached files.

**Do you want your identity to be public for this peer review?** For information about this choice, including consent withdrawal, please see our Privacy Policy .

Reviewer #1: No

---

## [Author Response · Author response to Decision Letter 1]

10 Jul 2025

We thank the reviewer and editor for their valuable comments that for sure helped us in further improving the quality of our manuscript.

-- Amendments based on Editor’s review: --

We carefully checked PLOS ONE's style requirements and applied them to our manuscript.

2. Please state what role the funders took in the study.

We adapted our financial disclosure according to the journal’s standards. The online submission form can be changed by using the following text:

“European Union’s Horizon 2020 research and innovation program under Marie Skłodowska-Curie (grant agreement No 860635). The funders had no role in study design, data collection and analysis, decision to publish, or preparation of the manuscript."

3. Please remove any funding-related text from the manuscript and let us know how you would like to update your Funding Statement.

We adapted the Acknowledgments Section of the manuscript according to the journal’s standards. To change the online submission form, please use the following text:

“Acknowledgments:

We thank the Erasmus MC iPS Core Facility for their contributions to this research. Specifically, we acknowledge Mehrnaz Ghazvini, PhD; Lieke Dons and Tracy Li for assisting with the cell culture experimental setup, and for their invaluable expertise.”

4. We strongly recommend all authors decide on a data sharing plan before acceptance, as the process can be lengthy and hold up publication timelines.

Relevant data are within the manuscript and its Supporting Information files. Additionally, we created an OSF Data Sharing Account and uploaded the gene expression data corresponding to the graphs presented in this article. Supporting information can be found in: https://osf.io/9fmb4/?view_only=c0cdb683481a476bbf7975ab0bdd9d7c

5&6. Please amend the manuscript submission data (via Edit Submission) to include author Filipa M. Duarte. & Please amend your authorship list in your manuscript file to include author Ana Duarte.

I unified the writing of my name to Filipa M. Duarte, both in manuscript, submission platform and letters. For clarification, the first author’s full name is Ana Filipa Madancos Duarte. Thus, Ana Duarte and Filipa M. Duarte are the same person.

7. Please include captions for your Supporting Information files at the end of your manuscript, and update any in-text citations to match accordingly.

We provided captions for our Supporting Information files at the end of our manuscript, and updated in-text citations accordingly.

8. Please review your reference list to ensure that it is complete and correct.

We reviewed our reference list to ensure that it is complete and correct.

-- Amendments based on Reviewer's input--

1. Reviewer’s comments to the Author:

“In Table 1 of the paper, numerous variables are described, including days of differentiation, cell number, coating matrix, and differentiation cocktails. Do the authors believe that cell number alone is the key factor influencing NCSC differentiation, or have the potential contributions of the other variables been considered?”

Our answer:

We thank the reviewer for raising this valid point. Yes, we believe that the cell number is the key factor that influences NCSC differentiation. More specifically, it is the combination of the initial cell seeding number and the number of differentiation days that will lead to a cellular monolayer at the end of the differentiation period.

Certainly, other factors like the applied matrix (read: ability of the cells to adhere to the matrix) also will play a role, however, after analyzing multiple NCC induction protocols, we noted that the cell number that was reached after the differentiation period (creating a cellular monolayer and not a cellular multi-layer), turned out to be the determining factor for efficient NCSC differentiation.

In order to clarify in our manuscript that we are aware of other intervening factors, we reinforced this idea as follows:

- Page 8; Line 172-176 (in final manuscript) ‘’ NCSCs differentiation protocols from hiPSCs show differences in multiple, partly intertwingled, factors that exert an influence in achieving NC identity such as seeding densities, duration of differentiation, coating matrix, and growth factors (Table 1). Among these, the most significant difference we observed was the considerable variation in the initial cell seeding densities.’’

2. Reviewer’s comments to the Author:

‘’Figure 1 was performed using cells from a single donor, whereas Figure 2 and on includes PBMC-derived hiPSCs from two donors. Could the authors clarify why a second donor was not included in Figure 1 to strengthen the robustness and reproducibility of the findings?’’

Our answer:

We appreciate the reviewer’s insight and are happy to provide clarification. We started the experimental work using the single, immediately available for testing, fibroblast-derived hiPSC line from Donor 1. However, our goal was to develop a hiPSCs to NCC differentiation protocol that can be used as a tool to study neural crest development and neurocristopathies.

Noticing the strong effect of the initial seeding density to subsequent NCC differentiation, we further optimized the protocol using the fibroblast-derived hiPSCs. After identifying the optimal seeding condition (17,000 cells/cm² followed by an eight day differentiation period), we expanded our testing to hiPSC lines from PBMC origin. Specifically, we applied the protocol to PBMC-derived hiPSCs from two donors, using both the optimal and the highest seeding densities tested (the highest seeding density tested was also de seeding density recommended by the manufacturer of the STEMdiff Neural Crest differentiation Kit - Stem Cell Technologies): 17,000 cells/cm² and 200,000 cells/cm², respectively. We observed analogous outcomes in the PBMC-derived hiPSC lines (Donors 2 and 3) compared to the fibroblast-derived hiPSC line. However, due to the different cell origins, data from the three lines could not be pooled together.

3. Reviewer’s comments to the Author:

‘’It is unclear what additional insight Figure 2 provides, as the downregulation of OCT4 and NANOG during differentiation is expected regardless of initial cell confluency. This trend was already demonstrated in Figure 1 using fibroblast-derived hiPSCs. Could the authors clarify the specific purpose of Figure 2?’’

Our answer:

We are thankful for the reviewer’s feedback and are happy to provide further clarification . We intend to maintain the experimental assessments in Donor 2 and Donor 3 identical to the work carried during optimization experiments conducted with the fibroblast-derived hiPSCs line from Donor 1. The purpose of this figure is to demonstrate that the PBMC-derived hiPSCs are indeed pluripotent cells, capable of expressing stemness genetic markers, as demonstrated in Donor 1 (Fig1B). Additionally, the figure demonstrates that once neural crest differentiation is induced, both lines successfully abandon their stemness state (downregulation of stemness markers OCT3/3 and NANOG). Certainly, a similar trend to what was observed in Figure 1. We appreciate the reviewer’s concerns, therefore, we turned this figure into supplementary Fig S1. This reinforces the results previously observed in fibroblast-derived hiPSCs and contributes to the robustness and replication of the protocol.

In order to clarify that the intent of this figure is to show replicable experimental procedure with identical observations, we reinforced this idea in the manuscript as follows:

- Page 11; Line 230-233 (in final manuscript): ‘’ To ensure consistency across experiments and validate the reproducibility of our observations, we designed the experimental setup to assess identical conditions, thereby minimizing any potential variability and confirming the reliability of the results observed in donor 1.’’

4. Reviewer’s comments to the Author:

‘’Please include statistical analysis for the qPCR data presented in Figure 1 and Figure 3 to support the reported differences in gene expression.’’

We are grateful for the reviewer’s suggestion. In order to report statistical analysis and significance, we included the following paragraph in the Methods section:

- Page 8; Line 164-167 (in final manuscript): ‘’Statistical analysis: A Mann–Whitney U test was used to assess the effects of cell seeding density and differentiation day on genetic marker expression. Statistical significance was set at p < 0.05 (*), p < 0.01 (), and p < 0.001 (*). Analyses were performed using Python.''

Additionally, in all Figures, except Figure 1, the significancy level was indicated wherever significance was reached. In Figure 1, a Figure loaded with information, for reasons of clarity, we chose to only indicate significance levels between the hiPSC and 17,000 cells/cm² bars and between the 17,000 cells/cm² and 200,000 cells/cm² bars – the optimal seeding density for NCC differentiation (17,000 cells/cm²) and maximum seeding density (200,000 cells/cm² recommended by the NCC differentiation kit’s guidelines).

5. Reviewer’s comments to the Author:

‘’For Figures 1C and 1D, the 17,000 cells/cm² condition could be highlighted more clearly to aid reader interpretation—perhaps by using connecting lines or another visual cue. Just a suggestion’’

We appreciate the reviewer’s suggestion and agree that a visual cue would improve the message of the manuscript. Therefore, in order to emphasize the optimal density of 17,000 cells/cm², as per their suggestion we underlined with a black stroke the 17,000 cells/cm² condition in each graph. A visual demonstration of one of the 6 graphs that were changed follows:

6. Reviewer’s comments to the Author:

‘’Line 237 – there is a repetition on ‘confluency’

Line 241 – ‘ain’’’

We are grateful for the reviewer’s attentive analysis of our manuscript and have proceeded with the corrections of the semantic errors (line numbers can be found in the resubmitted manuscript):

- Page 12; Line 254-255 (in final manuscript): ‘NC induction of hiPSCs seeded at 200,000 cells/cm² reached confluency as early as Day 1 of the differentiation period (Figs 2A and 2B).’’

- Page 12; Line 257-258 (in final manuscript): ‘’… which resulted in super confluency and the formation of multi-layered cell cultures over subsequent days.’’

7. Reviewer’s comments to the Author:

‘’In the legend for Figure 3, consider revising the title to: "Increased differentiation efficiency of PBMC-derived hiPSCs into neural crest stem cells at lower cell seeding densities." The same suggestion applies to the legend for Figure 4 to ensure clarity and consistency regarding the cell source’’

We appreciate the reviewer’s suggestion and agree that the title proposed for figure three is more clear and more consistent. Thus, we have changed the title of:

- Figure 2 legend; Page 13; Line 272-273 (in final manuscript): " Increased differentiation efficiency of PBMC-derived hiPSCs into neural crest stem cells at lower cell seeding densities."

- Figure 3 legend; Page 14; Line 307-308 (in final manuscript): ‘’ Neural crest differentiated PBMC-derived hiPSCs express neural crest markers in areas with a monolayer of cells and neuroectoderm markers in areas of multi-cell layers.’’

---

## [Decision Letter · Decision Letter 1]

11 Aug 2025

Reaching a cell monolayer at the end of hiPSC differentiation enhances neural crest lineage commitment.

PONE-D-25-17819R1

Dear Dr. M. Duarte,

We’re pleased to inform you that your manuscript has been judged scientifically suitable for publication and will be formally accepted for publication once it meets all outstanding technical requirements.

Kind regards,

Michael Klymkowsky, Ph.D.

Academic Editor

PLOS ONE

Additional Editor Comments (optional):

Reviewers' comments:

Reviewer's Responses to Questions

**Comments to the Author**

1. If the authors have adequately addressed your comments raised in a previous round of review and you feel that this manuscript is now acceptable for publication, you may indicate that here to bypass the “Comments to the Author” section, enter your conflict of interest statement in the “Confidential to Editor” section, and submit your "Accept" recommendation.

Reviewer #1: All comments have been addressed

2. Is the manuscript technically sound, and do the data support the conclusions?

Reviewer #1: Yes

3. Has the statistical analysis been performed appropriately and rigorously? 

Reviewer #1: Yes

4. Have the authors made all data underlying the findings in their manuscript fully available?

Reviewer #1: Yes

5. Is the manuscript presented in an intelligible fashion and written in standard English?

Reviewer #1: Yes

6. Review Comments to the Author

Reviewer #1: (No Response)

7. PLOS authors have the option to publish the peer review history of their article (what does this mean? ). If published, this will include your full peer review and any attached files.

**Do you want your identity to be public for this peer review?** For information about this choice, including consent withdrawal, please see our Privacy Policy .

Reviewer #1: No

---

## [Editor Report · Acceptance letter]

PONE-D-25-17819R1

PLOS ONE

Dear Dr. M. Duarte,

I'm pleased to inform you that your manuscript has been deemed suitable for publication in PLOS ONE. Congratulations! Your manuscript is now being handed over to our production team.

Kind regards,

on behalf of

Dr. Michael Klymkowsky

Academic Editor

PLOS ONE